# Encapsulation of Hydrophobic Porphyrins into Biocompatible Nanoparticles: An Easy Way to Benefit of Their Two-Photon Phototherapeutic Effect without Hydrophilic Functionalization

**DOI:** 10.3390/cancers14102358

**Published:** 2022-05-10

**Authors:** Limiao Shi, Christophe Nguyen, Morgane Daurat, Nicolas Richy, Corentin Gauthier, Estelle Rebecq, Magali Gary-Bobo, Sandrine Cammas-Marion, Olivier Mongin, Christine O. Paul-Roth, Frédéric Paul

**Affiliations:** 1Univ Rennes, INSA Rennes, ENSCR, CNRS, ISCR (Institut des Sciences Chimiques de Rennes)-UMR 6226, F-35000 Rennes, France; lmshi09@yahoo.com (L.S.); nicolas.richy@univ-rennes1.fr (N.R.); olivier.mongin@univ-rennes1.fr (O.M.); 2IBMM, Univ Montpellier, CNRS, ENSCM, 34293 Montpellier, France; christophe.nguyen@umontpellier.fr (C.N.); magali.gary-bobo@inserm.fr (M.G.-B.); 3NanoMedSyn, 15 Avenue Charles Flahault, 34090 Montpellier, France; m.daurat@nanomedsyn.com (M.D.); c.gauthier@nanomedsyn.com (C.G.); erebecq@gmail.com (E.R.); 4INSERM, INRAE, Univ Rennes, Institut NUMECAN (Nutrition Metabolisms and Cancer) UMR A1341, UMR S1241, F-35000 Rennes, France

**Keywords:** biocompatible nanoparticles, two-photon photodynamic therapy, star-shaped porphyrins, fluorescence bio-imaging, hydrophobic photosensitizers

## Abstract

**Simple Summary:**

Efficient photosensitizers for photodyanmic therapy (PDT) need to be soluble in physiologic media. This requirement often complicates significantly the chemical access to such compounds, resulting in lower availability and higher production costs for the best representatives. Given that their screening and selection is often initially conducted in organic media from series of hydrophobic model compounds, the possibility to use directly such hydrophobic photosensitizers in real PDT studies was highly desirable to speed up their definitive identification but also to alleviate their cost. In this respect, PMLABe polymeric nanoparticles (NPs) were presently probed as nanocarriers to water-solubilize hydophobic star-shaped porphyrin-based which turned out to be promising oxygen photosensitizers for theranostic approaches. We show here that PDT conducted using such NPs loaded with these compounds is as efficient than when functional hydrosoluble analogues of these photosensitiers are tested alone and that tracking of the photosensitizer by fluorescence imaging is even easier.

**Abstract:**

Star-shaped hydrophobic porphyrins, acting as powerful fluorescent two-photon photosensitizers for oxygen in organic solvents, can easily be loaded into PMLABe polymeric nanoparticles at various concentrations. In this contribution, the performance of these porphyrin-containing nanoparticles in terms of photodynamic therapy (PDT) is compared to those of the corresponding water-soluble porphyrin analogues when irradiated in MCF-7 cancer cells. While quite promising results are obtained for performing PDT with these nanoparticles, validating this approach as a mean for using more easily accessible and less expensive photosensitizers, from a synthetic perspective, we also show that their luminescence can still be used for bioimaging purposes in spite of their confinement in the nanoparticles, validating also the use of these nano-objects for theranostic purposes.

## 1. Introduction

Tetrapyrrolic macrocycles have attracted quite a lot of attention for cancer therapy since it was realized that these compounds were usually able to efficiently photosensitize oxygen upon visible light irradiation [1,2,3]. Accordingly, various types of porphyrins [4] or phthalocyanines [5,6] are currently used as photosensitizers for curing selected types of cancers via photodynamic therapy (PDT), especially these giving rise to small and localized tumors [7]. With the commercial photosensitizers, one-photon excitation is usually used to generate singlet oxygen in situ, leading to cell death once the photosensitizer has been properly internalized in the cancerous tissues. Current challenges in this field encompass the design of efficient emissive photosensitizers that might be excited via two-photon absorption (2PA) [8,9,10] since this kind of excitation has definite advantages over one-photon absorption (1PA), such as deeper penetration of the light beam used for photoexcitation in the living tissues, exquisite spatial control of the sensitized area, limited collateral tissular photodamages, and minimal autofluorescence from background [10,11]. In addition, when the photosensitizers are emissive, spectroscopy can also be advantageously used for localizing them in the tissular region under treatment (i.e., performing so-called «theranostics») [12,13].

We were quite active in this field these last years and have identified several promising porphyrin [14,15] or phthalocyanine-based [16] molecular architectures allowing to efficiently photosensitize oxygen while remaining sufficiently fluorescent for allowing imaging in organic solvents. However, the design of efficient photosensitizers also requires that they will be quantitatively internalized into cancerous cells. In this respect, we have used the quite common approach consisting in functionalizing the most active photosensitizers by short poly(ethylene glycol) (PEG) chains (TEG: triethyleneglycol), in order to increase their solubility in water [17,18]. Subsequent in vitro studies have revealed that, accordingly with our hopes, the new so-modified derivatives were biocompatible and constituted efficient photosensitizers for PDT upon two-photon excitation at 790 nm [17] or 840 nm [18]. Under such conditions, the viability of cancer cells was drastically reduced, while no toxicity was observed in the absence of irradiation. In addition, their sizeable emission in water allowed for their localization in the living tissues via fluorescence imaging, making them useful for theranostic approaches. Porphyrin derivatives such as **1a** and **2a** were thus among the best derivatives of that kind tested so far with MCF-7 breast cancer cells (Figure 1) [17].

However, their butylated parents **1b** [14] and **2b** [15], in spite of very similar optical performances for two-photon oxygen photosensitization (2P-PDT) and two-photon fluorescence imaging in organic solvents (2PEF; see ESI, Appendix A), cannot be used for biomedical applications in physiologic media because of their strong hydrophobicity. Considering that the synthesis of the water-soluble porphyrins **1a** and **2a** [17] requires a much more challenging multi-step protocol than that used for accessing **1b** and **2b** and that this feature significantly limits the availability (and cost) of these remarkable photosensitizers, a much more promising approach from a practical standpoint would be to directly use **1b** and **2b** for PDT instead. We recently became aware that this might be achieved by encapsulating the hydrophobic porphyrins **1b** and **2b** in biocompatible nanoparticles (NPs) formulated from either hydrophobic or amphiphilic poly(malic acid) (PMLA) derivatives [19,20,21]. Indeed, some of us have recently shown that NPs prepared from hydrophobic or amphiphilic PMLA derivatives can efficiently vectorize fully hydrophobic molecules into cells and that the resulting NPs suspensions present usually no significant in vitro and in vivo toxicity [19,20,21]. Furthermore, the surface of these NPs can be easily modulated by changing the type of PMLA derivatives used for their formulation, potentially leading to a range of NPs with different properties [22]. It is thus the purpose of this work to attempt loading NPs prepared from a hydrophobic PMLA derivative, the pol(benzyl malate) (PMLABe_73_), with **1b** and **2b** and to test their efficacy for PDT with MCF-7 breast cancer cells.

## 2. Results and Discussion

### 2.1. Preparation and Characterization of Porphyrin-Loaded NPs Based on PMLABe_73_

As shown on Figure 2, the first step toward PMLABe-based nanoparticles consists in the synthesis of the polymer precursor (PMLABe_73_) obtained by anionic ring opening polymerization of benzyl malolactonate (MLABe), prepared in four steps from aspartic acid (ESI) [20,21,23,24]. This polymerization is now well-mastered and leads to reproducible results in terms of structure and dispersity (Figure 2). The molar mass of the resulting PMLABe polymer is controlled by the ratio monomer/initiator, and fixed, for the present study, at 15,000 g·mol^−1^ (i.e., corresponding to 73 repeating units). ^1^H NMR and size exclusion chromatography analyses were in good agreement with this structure [19,20,21,23].

Starting from this well-defined hydrophobic homopolymer, the desired nanoparticles (NPs) were prepared by nanoprecipitation, a technique first described by Thioune et al. [24] This simple and reproducible method consists in a rapid addition of a water-miscible organic solvent containing both the polymer (PMLABe_73_) and the hydrophobic molecules of interest, into water under vigorous stirring. As a result of their hydrophobicity, the polymeric materials spontaneously aggregate to form NPs while the molecules of interest are entrapped into the hydrophobic inner-core of these NPs. After removing the organic solvent in vacuo, a stable suspension of NPs is usually obtained. In particular, hydrophobic homopolymers such as PMLABe_73_ lead to roughly spherical NPs with hydrodynamic diameters generally comprised between 100 and 300 nm depending on the conditions used [23]. In the context of the present study, we have evaluated the possibilities to encapsulate the two hydrophobic porphyrins of interest (**1b** and **2b**) into such NPs using different initial contents of porphyrins (10 and 1 wt% relative to the polymer mass). After elimination of non-encapsulated porphyrins by filtration through a Sephadex G25 column, the desired NPs suspensions were obtained and characterized. Dynamic light scattering (DLS) was used to determine their hydrodynamic diameters (D_h_) and dispersities (PDI), while transmission electron microscopy (TEM) was used to analyze their morphology. 

Considering the DLS measurements performed directly on the initial suspensions without any dilution (Table 1), the hydrodynamic diameters found for PMLABe_73_-based NPs were slightly above 200 nm, regardless of if the NPs were empty or porphyrins-loaded. Obviously, the loading of porphyrins (whatever their nature and the nature/amount of porphyrin encapsulated) does not significantly modify their hydrodynamic diameter nor their dispersity. Such porphyrin-loaded NPs can be considered as stable over time when kept at 4 °C. Indeed, we have previously shown that both diameters and dispersity of hydrophobic molecule-loaded PMLABe_73_-based nanoparticles did not significantly change upon incubation at 4 °C [25]. In order to evaluate if the presence of porphyrins inside has an impact on the morphology of these NPs, all the batches were also analyzed by transmission electronic microscopy (TEM). As shown in Figure 1, porphyrin-loaded NPs have a more or less spherical shape, as observed for empty PMLABe_73_-based NPs [12]. As already observed, the sizes determined from TEM images were slightly lower than those measured by DLS (see Appendix A; ESI) [21]. However, what is important to note is that encapsulation of porphyrins such as **1b** or **2b** can be achieved at various concentrations and has no significant influence on their morphology, regardless of the nature of the porphyrin encapsulated.

These initial suspensions were then concentrated to obtain NP concentrations compatible with in vitro assays. As described in the Experimental Section, all NP suspensions were ultra-centrifugated and filtrated through micro-con devices, and diluted to reach polymer concentrations of 5 mg·mL^−1^. 

The resulting NP suspensions were then again analyzed by DLS to check if this protocol did not alter their hydrodynamic diameter and dispersity (Table 2). The values contained in Table 1 and Table 2 again reveal that the ultra-centrifugation/filtration treatment does not strongly affect their hydrodynamic diameters in the suspensions. However, a slight overall increase of NP suspension dispersity (PDI) is observed which probably results from the formation of some aggregates at higher concentrations. Nevertheless, such changes should not affect significantly the properties of NP suspensions nor question their use for in vitro assays.

### 2.2. Emission Properties of the Porphyrin-Loaded NPs

All NPs exhibit UV-vis and emission spectra characteristic of the porphyrins in water, as demonstrated by the comparison with their analogues bearing triethyleneglycol (TEG) water-solubilizing chains (Figure 2 and ESI). The fluorescence quantum yields of the two types of NPs are comparable and have values comprised between 1% and 15% (Table 3). Interestingly, the NPs containing the porphyrins are even slightly more fluorescent (with quantum yields comprised between 3% and 15%) than the corresponding porphyrins with TEG chains (quantum yields around 3% in water) [17]. More precisely, we observe that NPs loaded at 1 wt% systematically exhibit higher quantum yields than corresponding NPs loaded at 10 wt% (by approximately a factor of two). Such an enhancement of the fluorescence quantum yield of a given fluorophore upon encapsulation in a polymeric environment was not expected [26] based on common observations [27] but has precedence in the literature [28].

To evaluate the actual concentration of porphyrin encapsulated in the NPs, the porphyrins were first fully released in solution by addition of THF to the water suspensions of NPs. After determining the molar extinction coefficients of porphyrins **1b** and **2b** alone dissolved in [90:10] THF/water mixtures (ESI), the absorption spectra of porphyrins released from NPs in the same [90:10] THF/water mixture were measured, allowing to calculate the porphyrin concentrations and finally to derive the experimental porphyrin loadings in NPs (Table 4). 

These experimentally derived concentrations are always lower than the theoretical expectations based on the concentration used during their synthesis. Overall, the yield of porphyrin incorporation relative to the theoretical value (%_exp_/%_theo_) varies from 46% to 77%. This difference possibly depends on the structure of the incorporated porphyrin and of the hosting polymer.

### 2.3. In Vitro PDT Assays Using Porphyrin-Loaded NPs

The NPs suspensions were vortexed just before testing in vitro. Living breast cancer cells (MCF-7) were incubated with NPs (25 µg·mL^−1^) for 20 h and two-photon excited fluorescence imaging (2PEF) was performed with a confocal microscope upon excitation at 790 nm with a *Chameleon* femtosecond laser. Cell membranes were stained with CellMask™ Orange (λ_ex_ = 561 nm). The presence of porphyrin-loaded NPs within the cells is clearly established for **NP1-4** (Figure 3, top), indicating that two-photon imaging is effective using NPs made from the PMLABe_73_ polymer. Notably, the empty NPs (**NP5**) are not fluorescent. To properly compare the imaging potential of these NPs loaded with butylated porphyrins with that of their water-soluble TEG-ylated analogues **1a** and **2a**, MCF-7 cells were also incubated with **1a** and **2a** at the very same concentrations as **1b** and **2b** embedded in **NP1-4** (Table 5), and 2PEF was performed under the same conditions. At these concentrations, the TEG-ylated porphyrins (Figure 3, bottom) appear much less bright than the corresponding porphyrin-loaded NPs.

Next, the potential of the NPs for two-photon photodynamic therapy (2P-PDT) was tested, by irradiating at 790 nm MCF-7 cells incubated with NPs (3 scans of 1.57 s at the maximum power laser of the *Chameleon* lamp with 3W laser input, using magnification ×10) and the surviving cells were quantified by a colorimetric assay 48 h after (Figure 4A). 

The first statement is that these NPs are non-toxic without light irradiation, regardless the presence of porphyrin inside, while a relatively effective 2P-PDT can be evidenced with NPs loaded with 10 wt% porphyrin upon irradiation at 790 nm. When NPs were replaced by PEG-ylated porphyrins at the same concentrations, no PDT effect was observed (Figure 4B). Additionally, according to these results, 2P-PDT with **NP3** seems more effective than with **NP1**, suggesting that **2b** acts as a better photosensitizer than **1b** in these NPs, in accordance with observations previously made on their water-soluble analogues **1a** and **2a** although these were used at much higher concentrations (25 µg·mL^−1^) [17].

Next, the 2P-PDT potential of the NPs was tested, by irradiating at 790 nm MCF-7 cells incubated with NPs (3 scans of 1.57 s at the maximum power laser of the *Chameleon* lamp with 3W laser input, using magnification ×10) and the surviving cells were quantified by a colorimetric assay 48 h after (Figure 4A). 

The first statement is that these NPs are non-toxic without light irradiation, regardless the presence of porphyrin inside, while a relatively effective 2P-PDT can be evidenced with NPs loaded with 10% porphyrin upon irradiation at 790 nm. When NPs were replaced by PEG-ylated porphyrins at the same concentrations, no PDT effect was observed (Figure 4B). Additionally, according to these results, 2P-PDT with **NP3** seems more effective than with **NP1**, suggesting that **2b** acts as a better photosensitizer than **1b** in these NPs, in accordance with observations previously made on their water-soluble analogues **1a** and **2a** when used at much higher concentrations (25 µg/mL) [17].

The efficiency of **NP1** and **NP3** in 2P-PDT was confirmed by the formation of reactive oxygen species (ROS) during the two-photon irradiation. Indeed, when incubated with **NP1** or **NP3** and irradiated in the same experimental conditions that those used for 2P-PDT, the cells (previously incubated with DCF-DA) exhibited an important green luminescence diagnostic of ROS generation, subsequent to transformation of the DCF-DA into fluorescent DCF (Figure 5). In contrast, based on luminescence, the TEG-ylated porphyrins did not induce any production of ROS, in line with the absence of any biological effect during the two-photon irradiation. 

In addition, the integrity of **NP1** and **NP3** was verified 10, 60, and 90 min after irradiation. For this, confocal imaging was performed on cells incubated 20 h with **NP1**, **NP3**, **1a**, **2a** and stained with *CellMask*. First, a picture was collected at time zero (×60 magnification), then cells were irradiated using 2P-PDT conditions and imaged (×10 magnification; 3 scans of 1.57 s). Finally, cells were imaged after 10, 60, or 90 min (×60 magnification) to check if NPs and TEG-ylated porphyrins were still intact. By this mean, approximately the same profile of bio-availability and localization inside the cells could be demonstrated during all the experiment after irradiation, for NP and TEG-ylated porphyrins (Appendix A). Only the data gathered after 60 min are shown but similar results were obtained at other times.

Then, to learn more about the action mode of these photosensitizers, we have studied the subcellular localization of **NP1** and **NP3** along with that of their free TEG-ylated porphyrins analogues in the cells. As expected, NPs were not localized in mitochondria (Figure 6A) but are (at least in part) localized in lysosomes (Figure 6B), as demonstrated by the yellow staining in merged pictures induced by the overlap of the red color of lysosomes and green color of NP (mainly **NP3**). This suggests an internalization via the endo-lysosomal pathway while their TEG-ylated porphyrin analogues **1a** and **2a** remain aggregated and difficult to locate. 

Finally, and in order to verify the possible use of such NPs with other cancer cells, their internalization was studied on highly aggressive cell lines such as MDA-MB-231 breast cancer (triple negative) and Capan-1 pancreas cancer cells. For this, confocal imaging was performed on cells incubated 20 h with **NP1** and **NP3** (25 µg·mL^−1^) and equivalent concentration of TEG-ylated porphyrins **1a** and **2a**. Cell membranes were stained with *CellMask*. **NP1** and **NP3** were efficiently internalized in both cancer cell lines (Figure 7), showing a strong bioavailability at the contrary of their TEG-ylated porphyrins (**1a** and **2a**). This result suggests an efficient targeting potential of such NP for various cancer cells. 

## 3. Materials and Methods

### 3.1. General

All commercial chemicals were used as received. Porphyrins **1a**, **2a** [17], **1b** [14], and **2b** [15] were synthesized according to published procedures.

### 3.2. Preparation of Starting Polymer

PMLABe_73_ was synthesized by anionic ring opening polymerization (aROP) of benzyl malolactonate (MLABe) in presence of tetraethylammonium benzoate [NEt_4_^+^,C_6_H_5_COO^−^] as initiator [19,23]. The molar mass of PMLABe was fixed at 15,000 g·mol^−1^ (polymerization degree = 73) by the ratio MLABe/initiator. After purification by precipitation and characterization by nuclear magnetic resonance (NMR) and size exclusion chromatography (SEC), the hydrophobic homopolymer (PMLABe_73_) was used to prepare porphyrin-loaded NPs using nanoprecipitation, a simple and reproducible technique [24], previously described to prepare PMLABe-based NPs loaded or not with a hydrophobic molecule of interest [19,20,21].

The two hydrophobic porphyrins of interest (**1b** and **2b**) were then encapsulated into PMLABe_73_-based NPs using two different amounts (10 and 1 wt% relative to the mass of polymer).

### 3.3. Preparation of Porphyrin Stock Solutions

Stock solutions of porphyrins **1b** and **2b** in THF were prepared at a concentration of 1 mg·mL^−1^.

### 3.4. Preparation of PMLABe_73_-Based NPs

***PMLABe_73_[1b] 10 wt% NPs (NP1):*** 5 mg of PMLABe_73_ was dissolved into 500 µL of THF and 500 μL of the red solution of **1b** in THF was added (10 wt% relative to the polymer, i.e., 0.5 mg of **1b**). This solution was quickly added to 2 mL of water under vigorous stirring. The milky red/brown mixture was stirred at room temperature for 10 min. THF was then evaporated under vacuum using a rotary evaporator. The vacuum was lowered to 80 mbar and maintained at this pressure for 10 min. After evaporation of the organic solvent, traces of precipitate were visible on the flask walls. The volume of the final solution was then adjusted to 2 mL by adding water and the solution was deposited on the top of a Sephadex column. When all the sample had entered the column, 0.5 mL of water was added, followed by further addition of 3.5 mL (after elution, no trace of free **1b** was visible on the column). The NPs loaded with **1b** were collected in a vial (final volume ≈ 3.5 mL). The suspension was analyzed by DLS (Table 1) and TEM (Figure 1). The NPs suspension was first placed in the filters of micro-cons and was ultra-centrifuged/filtered at 15,000× *g* for 5 min (Molecular Weight Cut Off (MWCO) = 10 kDa). The filters were then inverted and the solution was centrifuged at 1000× *g* for 1 min. The recovered suspension containing the title NPs was then diluted in water to a total volume of 1 mL, i.e., corresponding to a polymer concentration of 5 mg·mL^−1^ before being analyzed by DLS (Table 2).

***PMLABe_73_[1b] 1 wt% NPs (NP2):*** 5 mg of PMLABe_73_ was dissolved into 950 µL of THF and 50 μL of the red solution of **1b** in THF was added (1 wt% relative to the polymer, i.e., 0.05 mg of **1b**). This pinkish solution was quickly added to 2 mL of water under vigorous stirring. The milky pale-yellow mixture was stirred at room temperature for 10 min. The THF was evaporated as described above for NP1. The volume of the final solution was adjusted to 2 mL by adding the necessary amount of water and the solution was deposited on a Sephadex column and eluted as described above for NP1 (after elution, no trace of free **1b** was visible on the column). The solution containing the NPs loaded with **1b** was collected in a vial (final volume ≈ 3.5 mL) and analyzed by DLS (Table 1) and TEM (Figure 1). In order to obtain a polymer concentration of 5 mg·mL^−1^, this suspension was ultra-centrifuged/filtered on micro-con systems (MWCO = 10 kDa), as described for NP1. The resulting suspension containing the title NPs was analyzed by DLS (Table 2).

***PMLABe_73_[2b] 10 wt% NPs (NP3):*** 5 mg of PMLABe_73_ was dissolved into 500 µL of THF and 500 μL of the red solution of **2b** in THF was added (10 wt% relative to the polymer, i.e., 0.5 mg of **2b**). This solution was quickly added to 2 mL of water under vigorous stirring, giving a milky red/brown mixture which was stirred at room temperature for 10 min. The organic solvent was then evaporated as described above for NP1 and diluted to 2 mL by adding the necessary amount of water. The NPs were then purified on a Sephadex column as previously described for NP1 and the resulting suspension of title NPs was analyzed by DLS (Table 1) and TEM (Figure 1). In order to obtain a polymer concentration of 5 mg·mL^−1^, this suspension was ultra-centrifuged/filtered on micro-con systems (MWCO = 10 kDa), as described for NP1. The resulting suspension containing the title NPs was analyzed by DLS (Table 2).

***PMLABe_73_[2b] 1 wt% NPs (NP4):*** 5 mg of PMLABe_73_ was dissolved into 950 µL of THF and 50 μL of the red solution of **2b** in THF was added (1 wt% relative to the polymer, i.e., 0.05 mg of **2b**). This pinkish solution was quickly added to 2 mL of water under vigorous stirring, giving a milky pale-yellow mixture which was stirred at room temperature for 10 min. The organic solvent was then evaporated as previously described for NP1 and diluted to 2 mL by adding the necessary amount of water. The NPs were then purified on a Sephadex column as described for NP1 and the resulting suspension of title NPs was analyzed by DLS (Table 1) and TEM (Figure 1). In order to obtain a polymer concentration of 5 mg·mL^−1^, this suspension was ultra-centrifuged/filtered on micro-con systems (MWCO = 10 kDa), as described for NP1. The resulting suspension containing the title NPs was analyzed by DLS (Table 2).

***PMLABe_73_ NPs (NP5):*** 5 mg of PMLABe_73_ was dissolved into 1 mL of THF. This solution was quickly added to 2 mL of water under vigorous stirring and the resulting milky mixture was stirred at room temperature for 10 min. The THF was then evaporated as described above for NP1 and the volume of the final solution was adjusted to 2 mL by adding water. The NPs were then purified on a Sephadex column as described for NP1 and the resulting suspension of title NPs was analyzed by DLS (Table 1) and TEM (Figure 1). In order to obtain a polymer concentration of 5 mg·mL^−1^, this suspension was ultra-centrifuged/filtered on micro-con systems as described for NP1. The resulting suspension containing the title NPs was analyzed by DLS (Table 2).

### 3.5. Cell Culture

All cells used in this study were purchased from ATCC. They were cultivated in their appropriate culture medium (from Gibco^TM^) and allowed to grow at 37 °C, in humidified atmosphere, with 5% CO_2_. Human breast cancer cells MCF-7 were cultured in DMEM F12-GlutaMAX™-I (containing 4.5 g·L^−1^ of D-glucose) + 10% fetal bovine serum (FBS) + 1% penicillin/streptomycin (P/S). Human breast cancer cells MDA-MB-231 were cultured in DMEM supplemented with 10% FBS and 0.5% gentamycin. Human pancreas cancer cells Capan-1 were cultured in DMEM-GlutaMAX™-I (containing 4.5 g·L^−1^ of D-glucose) + 10% FBS + 1% P/S.

### 3.6. Methods

***Dynamic light scattering (DLS):*** DLS measurements were performed on a *Nano-sizer ZS90* (Malvern) at 25 °C, with a He-Ne laser at 633 nm and a detection angle of 90°.

***Transmission electron microscopy (TEM):*** TEM images were recorded using a *Jeol 2100* microscope equipped with a *Glatan Orius 200D* camera, using a 80 KeV accelerating voltage on the THEMIS platform (ISCR—Rennes). Each sample was deposited on a *Formvar* carbon film coated on a 300-mesh copper grid. After 6 min, the excess of sample was removed and staining was performed with phosphotungstic acid (1% volume).

***Spectroscopic Measurements.*** All photophysical properties were performed with freshly prepared air-equilibrated solutions at room temperature. UV-Vis absorption spectra were recorded on a Jasco V-570 spectrophotometer. Fluorescence measurements were performed on dilute solutions (*ca.* 10^−6^ M, optical density <0.1) contained in standard 1 cm quartz cuvettes using an Edinburgh Instrument (FLS920) spectrometer in photon-counting mode. Fully corrected emission spectra were obtained, for each compound, after excitation at the wavelength of the absorption maximum, with *A_λ_*_ex_ < 0.1 to minimize internal absorption. Fluorescence quantum yields were measured according to reported procedures using tetraphenylporphyrin in toluene as a standard (quantum yield *ϕ*_F_ = 0.11) [29,30,31].

***Two-photon fluorescence imaging:*** Cells were seeded onto bottom glass dishes (*World Precision Instrument*, Stevenage, UK) at a density of 10^6^ cells.cm^−2^ in culture conditions previously established. The day after seeding, cells were incubated 20 h with the various NPs or with **1a** or **2a** at the determined concentration. Cell membranes were stained with Cell Mask orange (*Invitrogen*, Cergy Pontoise, France) at a final concentration of 5 μg·mL^−1^ added 15 min before the end of incubation, and imaged at 561 nm. Lysosomes and mitochondria were stained with *LysoTracker* and *MitoTracker* (*Invitrogen*, Cergy Pontoise, France) and revealed under excitation wavelength of 577 nm and 490 nm, respectively. Cells were incubated with *LysoTracker* at a final concentration of 1 µM added 2 h before the end of incubation or *MitoTracker* at a final concentration of 0.1 µM added 45 min before the end of incubation. Then, cells were washed twice with culture medium and fluorescence imaging was performed on living cells with a *LSM 780 LIVE* confocal microscope (*Carl Zeiss*, Le Pecq, France), at 790 nm for various NPs, **1a** and **2a**. Images were performed with high magnification (63x/1.4 OIL DIC Plan-Apo).

***Two-photon photodynamic therapy (TP-PDT):*** Cells were seeded into 384-well glass bottom at 500 cells per well in 50 µL of culture medium and allowed to grow for 24 h. Then, cells were incubated for 20 h with the various NPs, **1a** or **2a**. After incubation, living cells were submitted or not to a irradiation with a pulsed laser, performed with a *LSM 780 LIVE* confocal microscope (*Carl Zeiss*, Le Pecq, France), at 790 nm (10×/0.3) with a focused laser beam and a maximum laser power (3 W input, 900 mW·cm^−2^ output before the objective). Half of the well was irradiated at 790 nm by 3 scans of 1.57 s duration in four different areas of the well. The scan size does not allow irradiating more areas without overlapping. After 2 days, a cell death quantification assay was performed by incubating cells with thiazolyl blue tetrazolium bromide (MTT) (0.5 mg·mL^−1^) for 4 h to determine the mitochondrial enzyme activity. Then, supernatant was removed, and 150 µL of EtOH/DMSO (1:1) was added to dissolve the MTT precipitates. Absorbance was measured at 540 nm with a microplate reader. Because only half of the wells were irradiated here, the value obtained was corrected according to the following formula: Abs “no laser”—2 × (Abs “no laser”—Abs “laser”).

***Reactive oxygen species detection under two-photon excitation:*** The detection of intracellular reactive oxygen production (ROS) was performed using DCF-DA Cellular ROS Detection Assay Kit (abcam). Cells were seeded and treated with NP and porphyrins in the same conditions as those for TP-PDT excepted that, 45 min before irradiation, cells were incubated at 37 °C with 20 µM of H_2_DCF-DA (abcam). Then, cells were irradiated, rinsed twice with cell media, and the fluorescence emission (green) of DCF was detected at 535 nm, traducing the generation of ROS. 

***Statistical study:*** Statistical analyses were carried out using Student *t*-test, to compare paired groups of data (“Laser” vs. “No laser”). A *p*-value < 0.05 was considered to be statistically significant.

## 4. Conclusions

We have shown here that hydrophobic porphyrins such as **1b** and **2b** can actually be loaded at various concentrations in NPs made from the PMLABe polymers. The resulting NPs are biocompatible and, in line with the 2PEF and 2P-PDT performances of **1b** and **2b** in organic solvents (ESI, Appendix A), can be used for two-photon excited fluorescence imaging and two-photon photodynamic cancer therapy in water. Their fluorescence quantum yields in water are improved relative to those of the water-soluble porphyrins **1a** and **2a**, suggesting that the stacking of **1b** and **2b** in these NPs is beneficial to fluorescence imaging. Possibly, the confinement in the NPs prevents detrimental self-quenching process after two-photon excitation. While all these NPs appear non-toxic in absence of light, we show that some of them exhibit photosensitizing properties upon excitation in the near-IR range (790 nm). As a result, the NPs loaded with 10 wt% of either **1b** or **2b** are now effective at killing breast cancer cells at much lower concentrations than their TEG-ylated analogues **1a** and **2a**. Their efficiency and action mode (**NP1** and **NP3**) were also confirmed by detection of reactive oxygen species (ROS) during two-photon irradiation, whereas such ROS were absent in the case of their TEG-ylated analogues at the same concentrations. In this respect, NPs loaded with **2b** apparently give rise to more active photosensitizers than those loaded with **1b**. In MCF-7 cells, **NP1** and **NP3** are localized, at least in part, in lysosomes describing an internalization of NPs via the endolysosomal pathway. These NPs are also efficiently internalized in MDA-MB-231 breast cancer (triple negative) and Capan-1 pancreas cancer cells, illustrating their possible application with other cancer cells. Considering the large synthetic flexibility offered for surface functionalization by such PMLABe_73_-derived NPs, some room is left for enhancing the cell internalization of related NPs containing hydrophobic photosensitizers such as **1b** or **2b**. Furthermore, given the significantly simplified synthetic access to these derivatives compared to their TEG-ylated analogues **1a** or **2a**, this work opens very appealing perspectives for testing other hydrophobic star-shaped porphyrins, also synthetically much more accessible than **1a** or **2a**, as two-photon fluorescent photosensitizers for theranostic uses with various types of cancers presenting small and localized tumors.

## Data Availability

The data presented in this study are available in this article and Appendix A.

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
