# Peer review of "Encapsulation of Hydrophobic Porphyrins into Biocompatible Nanoparticles: An Easy Way to Benefit of Their Two-Photon Phototherapeutic Effect without Hydrophilic Functionalization"

_cancers, 2022, doi:10.3390/cancers14102358_

Round 1

Reviewer 1 Report

the authors' responses to the comments are reasonable and the manuscript is improved according to the comments from the reviewers. Thus, I think that this manuscript is now suitable for the publication in your journal.

Reviewer 2 Report

I approve the corrections provided in the revised paper and accept the interpretation of the authors.

Revised manuscript can be recommended for publication.

This manuscript is a resubmission of an earlier submission. The following is a list of the peer review reports and author responses from that submission.

Round 1

Reviewer 1 Report

The aim of the paper is clear. The manuscript is well organized. Most of the results are reliable, conclusions are supported by scientific evidences. However, the paper in its present form cannot be recommended for publication.

Major comments and questions

Spectroscopic instrumentation and experimental conditions for UV-VIS spectra emission spectra recording and quantum yield determination are not presented in Materials and Methods.

Molar extinction of free base porphyrins measured in solution is used for the concentration determination of NP incorporated porphyrins (line 174-177). Concentration determination based on this approach cannot be accepted. There is no reason to suppose that molar extinction in solution is equal to molar extinction after NP incorporation,  since absorption parameters are sensitively changing with the molecular environment.

What does “experimental uncertainty” mean? (line 144). Is it experimental error? Distribution width of Dh? Instrumental inaccuracy? How was it determined?

Minor comments

Contents of tables should be clearly specified in table legends.

Cells can be photosensitized not oxygen (line 48)

Reviewer 2 Report

In this manuscript, Limiao Shi et al synthesized porphyrins and nanoparticles encapsulating the porphyrins to demonstrate PDT effects on MCF-7 cells with two-photon excitation. The authors developed the hydrophobic porphyrins with or without TEG modifications. The hydrophobic ones without TEG linkers are encapsulated into a polymer (PMMLABe). The nanoparticles of the polymer and the porphyrin were characterized by use of TEM, DLS, and UV-Vis absorption and fluorescence spectrometer. Finally, the porphyrins with TEG modifications and the nanoparticles was added on MCF-7 cells and PDT effects were tested.

Although this manuscript includes interesting results, this reviewer thinks that more results are required for publication of this manuscript in the Cancers journal.

  1. Figure 4: The photoirradiation with 1a and 2a does not show significant effect on cell viability. The authors firstly need to optimize experimental conditions and procedures, involving cell treatment, photoirradiation, concentration of the molecules. Although the NP3 resulted in the decrement of cell viability, the results is not yet significant, by comparing previous reports for PDT on MCF-7 cells (for example, see https://doi.org/10.1038/s41467-018-04771-y and some references therein).

  1. Figure 3: Please discuss quantitatively cell penetration efficiency with or without encapsulation. Localization should be studied for the molecules and the nanoparticles.

  1. Please study if the porphyrins are released in cells before and after photoirradiation.

  1. It is required to confirm ROS generation efficiency with and without encapsulation.

  1. This reviewer is not sure, but the two-photon excitation could reduce the PDT effects on the cells. Please confirm with the normal (single photon) photoirradiation with wavelengths corresponding to the soret and Q bands to compare with the results of the two-photon PDT and ROS generation.